# Method of Sequential Approximation in Modelling the Processes of Heat Transfer and Gas Dynamics in Combustion Equipment

**Miroslav Rimar** [1], **Oleksandr Yeromin** [2], **Grigoriy Larionov** [3], **Andrii Kulikov** [1], **Marcel Fedak** [1],
**Tibor Krenicky** [4,*], **Olena Gupalo** [2] and **Yana Myanovskaya** [2]

[1]  Department of Process Technique, Faculty of Manufacturing Technologies, Technical University of Košice, Bayerova 1, 08001 Prešov, Slovakia

[2]  Institute of Industrial and Business Technologies, Ukrainian State University of Science and Technology, Lazaryana Str., 2, 49010 Dnipro, Ukraine

[3]  M.S. Poliakov Institute of Geotechnical Mechanics of the National Academy of Sciences of Ukraine, Simferopolska Str., 2a, 49005 Dnipro, Ukraine

[4]  Department of Technical Systems Design and Monitoring, Faculty of Manufacturing Technologies, Technical University of Košice, Bayerova 1, 08001 Prešov, Slovakia

*  Correspondence: tibor.krenicky@tuke.sk

**Abstract:** The behavior of the processes taking place in furnaces determines the efficiency of fuel chemical energy utilization, the quality of the final products and the environmental safety of the production. Mathematical models of the processes of gas dynamics and heat transfer in the working space of heating equipment are quite complex, and do not allow the establishment of a direct analytical relationship between the quality indicator of the process ($F$) and the influencing parameters ($x_i$). To simplify the procedure for obtaining the values of the function $F$ depending on the change of parameters $x_i$, a method of successive approximation is presented in the article. The main idea of the method is that the representation of the function around a point from the domain of the function can be extended to the entire domain for many problems of mechanics. The relative error in the definition of the function acquires its maximum value at the border of the area, and a reasonable narrowing of it allows control of the size of the error. Thus, the advantages of using the method are obvious; it is able to provide approximation of the function in a multiplicative form with a controlled error. The distribution of the method to the field of heat transfer problems is presented in this paper. The successful implementation of this method for solving problems of this kind shows that the solution of practical problems may be generalized for the entire domain of the function, despite the fact that the errors of such a representation increase to 5–7% when approaching its limit, which, however, may be considered acceptable for engineering calculations.

**Keywords:** gas dynamics; heat transfer; mathematical modelling; method of successive approximation; heating; standard of heating

## 1. Introduction

Material heating and heat treatment belong to fundamental processes that take place in furnaces heated by gaseous fuel. The processes of fuel combustion, movement of gases in the working chamber of the furnace, and external and internal heat exchange during heating have a high level of mutual interaction. Some parameters of these processes are uncertain and interdependent, and they impose numerous conditions of mutual application.

The main tool for examination of thermophysical processes in energetics, especially in metallurgy, is mathematical modelling. Owing of the high level of computer technology, the complexity of the software, and the ability to simulate the influence of different parameters, mathematical modelling prevails over real measurements. Modern mathematical models

that use numerical methods allow implementation of an algorithm of any complexity, as well as considering a large number of non-linear parameters affecting processes. However, the increase in the number of parameters that is considered and studied in the mathematical model significantly increases the amount of preparatory work for modelling, complicates software development and handling, requires the introduction of additional databases and data arrays, and increases the calculation time. Thus, for example, the calculation time of one of the options for heating a cage of twelve multi-ton ingots in a regenerative heating well using a mathematical model may result in more than 5 h of computation time [1]. It should be noted that the accuracy of such a calculation usually depends on the reliability of the accepted values of the properties of the materials, the environment influence, and other coefficients used in the model. As practice has shown, the results of the calculations may fully correspond to the real data collected during 5-year operation of the simulation object—a regenerative heating well [1].

Depending on the required accuracy of calculations, complex mathematical models of gas-dynamic, mass- and heat-transfer processes or more simplified engineering models are used as an engineering tool, that for particular cases provide sufficient accuracy of calculations. A similarly important tool provides the physical modelling of processes and aggregates, which enables a clear and easy-to-understand representation of the studied phenomena [2].

To ensure the accuracy of the calculations sufficient for the development of engineering solutions and the evaluation of proposed modernization measures, or to improve the design of the furnace and its equipment, it is quite sufficient to obtain the dependence of the change of the selected final criterion of the process (the so-called quality function $F$) on the factors (independent variable parameters $x_i$) that is influenced. This, of course, does not exclude the need to create a mathematical model (MM) of the processes under study as a perfect mechanism for a calculation experiment, but it allows significantly reduction of the number of necessary calculations without significantly losing the quality of the obtained result.

Multiplicative models are widely used to process the results of calculation experiments (obtaining the function $F = f(x_i)$).

However, the problem of choosing a multiplier that precedes the product of functions has not been fully clarified. This multiplier is also called the coefficient of ignorance, and the evaluation of the obtained data depends on its choice. Initially, the multiplier was chosen as a function of independent parameters from the domain. It turned out that despite the high accuracy and significant costs for conducting the experiment, the quality function of the process had satisfactory accuracy only around the reference point. Attempts to determine the quality function at other points did not yield a positive result, despite the use of various multiplier selection algorithms [3]. It became clear that an experiment carried out at a point simply cannot satisfy the accuracy of the definition of the quality function at other points and should not be a function of coordinates, but a number for each point in the domain.

On the other hand, any research process that is conducted using experiments, simulations, or mathematical models can be represented as a black box (Figure 1). Any process that can be represented in the form of a black box model can be applied to the method of successive approximation (SAP) used in this paper. Unlike many methods with a similar name, it does not use any derivatives of functions, which makes it attractive for use in a wide range of tasks.

The black box model may be represented here by physical devices, systems of differential equations, mathematical formulas, and computer programs or application software packages (ASPs) that implement it. The results of research using the MM, as a rule, represent matrices of numbers or tables of numerical data. Functions are reproduced using their values on the parameter grid. The finer the mesh of a model, the more accurate approximations of the function are achievable. For complex problems, obtaining the values of functions on the parameters mesh requires significant expenditure of machine time, which makes the problem of reproduction of functions practically impossible. Therefore, the idea of using the method of obtaining MM according to a simplified procedure arose.

According to this approach, the MM is obtained not on the parameters mesh, but at a certain point of the area of their definition. The algorithm proposed by the author of [4] makes it possible to significantly reduce the computational costs of reproducing the quality function, but only in the vicinity of a certain point. Thus, instead of calculating the values of the function on the parameter mesh, they are calculated only on the coordinate lines, which significantly reduces the number of calculations of the resulting function (Figure 2).

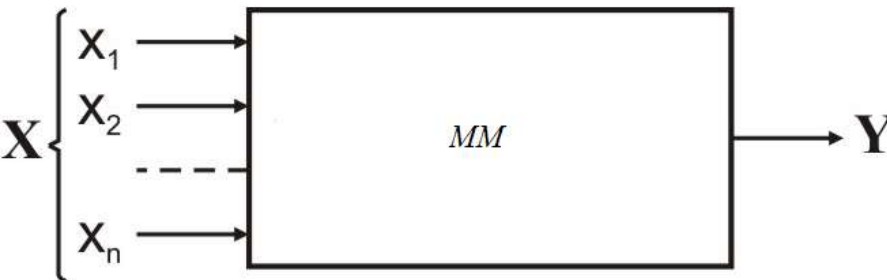

**Figure 1.** Model of a black box: $X(x_1, x_2, \ldots x_n)$, control parameters or model variables. Y is the resulting factor or process quality function $Y = Y(x_1, x_2, \ldots x_n)$, the form of which must be reproduced from the data obtained after computer processing.

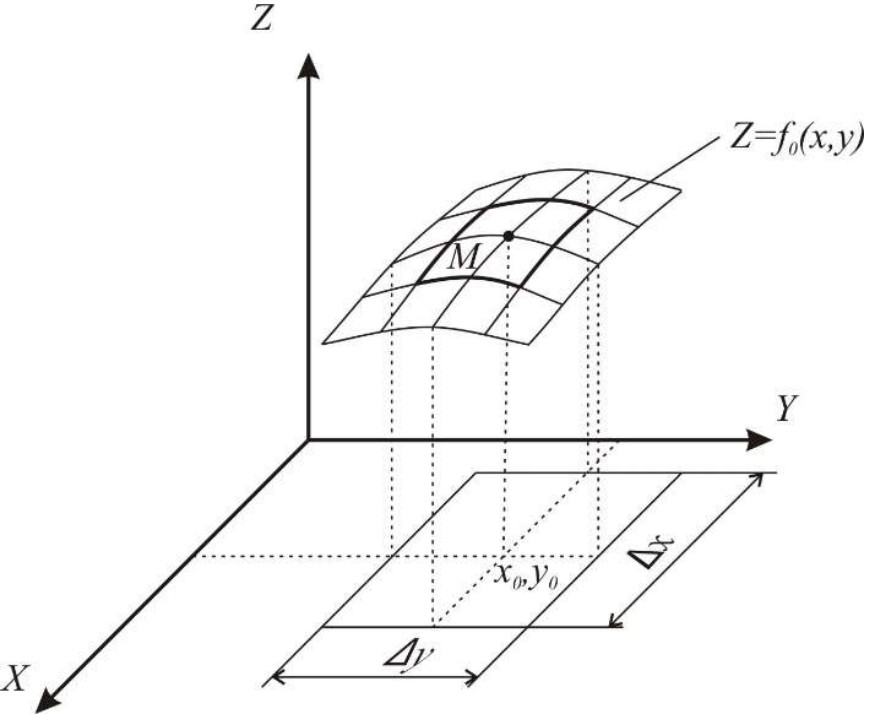

**Figure 2.** Selection of the point $M$ vicinity with coordinates $(x_0, y_0)$: $Z = f_0(x,y)$—the general view of the surface of the quality function, $\Delta x$, $\Delta y$—the step along the $x$ and $y$ coordinates.

The method of successive approximation (MSA) is a method that can be used in all fields from physical calculation to statistic and computer technology. It is possible to build an efficient and flexible computational method by using MSA [5]. This may be applied to the study of the fixed-point problems in the more abstract setting of Banach spaces (e.g., differential and integral equations, dynamical systems) [6]. The efficiency of such a method was also proven by comparing it to a variation iteration method under equivalent conditions [7].

MSA allows representation of functions in an analytical form (namely, in the form of a product of functions, each of which depends on one variable) when their values exist in

tabular form on lines formed by the intersection of the functional surface with coordinate planes passing through a selected point from the definition area (Figure 2) [8].

The examples of MSA successful application on the problems of geotechnical mechanics [4] show that the solution of practical problems may be extended to the entire domain of the function, despite the fact that the errors of such a representation increase when approaching its boundary, and do not exceed the value of 5–7%. It turns out that such accuracy is satisfactory for engineering calculations in the field of geotechnical mechanics since the initial data for this area are determined with the same accuracy.

The main aim of this work was to determine the possibility of applying the method of successive approximation for the study of gas dynamics and heat transfer processes occurring in furnaces during the heating of metal products, and to estimate the error of approximation of the quality function from independent process parameters.

## 2. Materials and Methods

### 2.1. Mathematical Model

A heating furnace with a U-shaped trajectory of flue gas movement was used to solve the problem, the mathematical model of gas dynamics and heat transfer processes [9]. The scheme of the furnace is shown in Figure 3. The furnace is designed for heating metal ingots (3) before pressure treatment. The working space (2) of the furnace has the shape of a rectangular parallelepiped. To burn fuel, the furnace is equipped with a burner (4) located in the upper part of the left side wall of the stove. In the lower part of the same wall, there is a window (5) for the exhaust of a flue gas from the working space of the furnace into the flue gas channel, equipped with regenerators (1) for heating the air used for burning fuel.

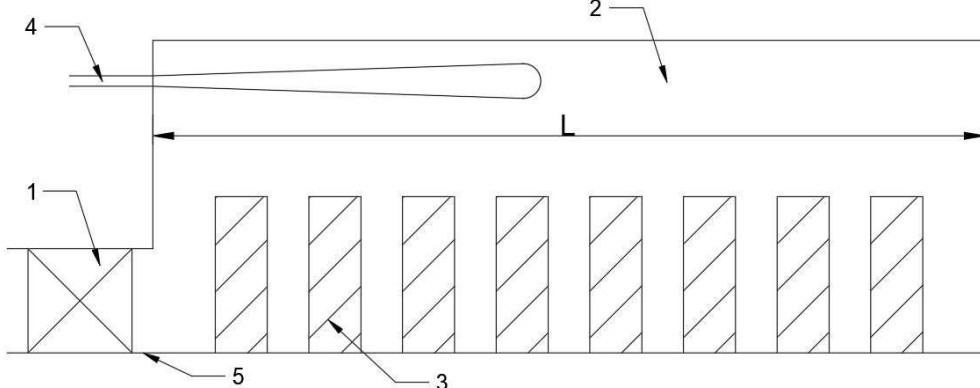

**Figure 3.** Scheme of a reheating furnace with regenerators and with distributed volumetric combustion: 1—regenerator; 2—working chamber; 3—heated material; 4—gas nozzle; 5—flue gas ducts; L—furnace length.

The furnace with a U-shaped trajectory of flue gas is among widely used equipment for material heating, so it was chosen to increase its parameters and, as a result, reduce of production costs and influence on the environment.

The main disadvantage of furnaces with a U-shaped trajectory of flue gas is the non-uniformity of gas temperatures in the working chamber, which does not allow obtaining the same quality of heating of all heated products [10]. Unevenness of heating in furnaces leads not only to an undesirable increase of the temperature of some product parts, but also to additional heat and material loss [11]. Therefore, the main tasks set during simulation of the processes taking place in such furnaces are:

- determination of gas temperature distribution along the trajectory of their movement in the working chamber of the furnace;
- calculation of the parameter of uniformity of gas temperature distribution in the working chamber ($F = \Delta t = t_{max} - t_{min}$);
- selection of independent parameters that affect uniformity;

- determining the necessary values of these parameters to increase the uniformity of gas temperature distribution in the working chamber.

The design parameters of the furnace and auxiliary equipment of the furnace are selected according to the results of mathematical modelling. For existing furnaces, the simulation results are used to develop measures to change the operating parameters of the unit and its equipment in order to improve the uniformity of gas and temperature distribution in the furnace.

During simulation, the working chamber of the furnace is divided into $i = 2n$ calculation zones (see Figure 4). It is assumed that the heating material is located in each calculated zone of the furnace, and there is an ideal mixing of gases; that is, the temperature of the gases within the volume is the same. A burner is located in the first (by gas flow) calculation zone $i = 1$, and a smoke window is located in the last zone $i = 2n$. The location of the burner and the smoke window on the same end wall contributes to the occurrence of large-scale gas recirculation, which significantly affects the uniformity of the smoke temperature distribution in the working chamber of the furnace.

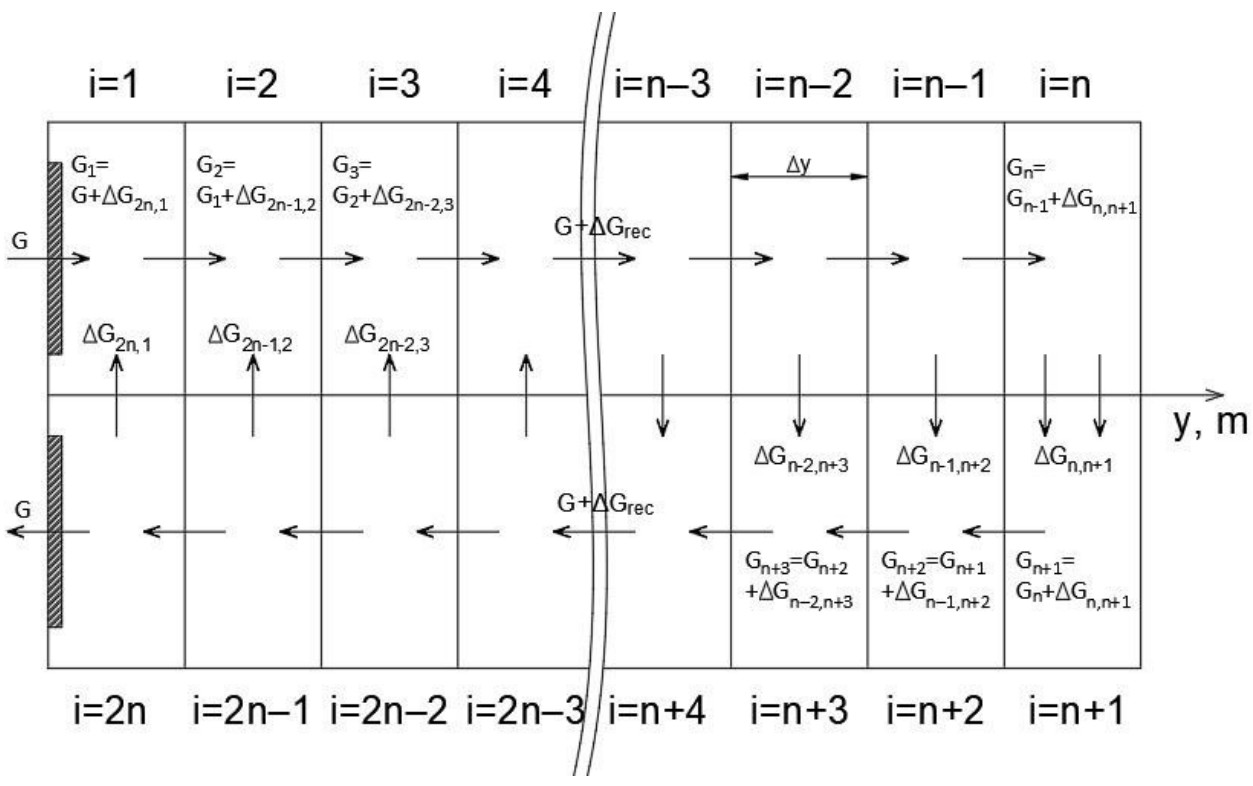

**Figure 4.** Calculation scheme of the furnace and the movement of flue gases.

In the central part of the furnace (Figure 4), on the border of the conditional division of the gas flow into forward and reverse trajectories of movement, local turbulent vortices appear, moving in forward and reverse directions. The mixing of gases in local zones through the flow separation boundary is a local recirculation of gases, which also affects the uniformity of the temperature distribution in the working chamber of the furnace.

The equation of heat balance of the U-shaped flue gas trajectory furnace is:

$$Q_{in\ i} + Q_{rec\ i} + Q_{bur\ i} = Q_{Me\ i} + Q_{lost\ i} + Q_{exg\ i} \tag{1}$$

where

$Q_{in\ i}$—heat entered into the 1st zone with fuel, air and fuel combustion products, (W);
$Q_{rec\ i}$—heat of combustion products passing through the furnace gas flow interface on the straight and reverse path, (W);

$Q_{bur\ i}$—heat released by the fuel combustion, (W);

$Q_{Me\ i}$—heat consumed to heat up the material in the zone, (W);

$Q_{lost\ i}$—heat loss through furnace walls, (W);

$Q_{exg\ i}$—heat loss of furnace gases leaving the $i$th zone, (W).

Heating components of the heat balance calculation are calculated according to Equations (2)–(4).

$$Q_{in\ \ i} = \begin{cases} B_f \cdot t_f \cdot c_f + L_n \cdot B_f \cdot t_{air} \cdot c_{air} & \text{for} \quad i = 1 \\ \left(1 - \sum\limits_{k=1}^{i-1} m_k\right) \cdot \left(B_f \cdot t_{i-1} \cdot c_f + L_n \cdot B_f \cdot t_{i-1} \cdot c_{air}\right) + \\ + \sum\limits_{k=1}^{i-1} m_k \cdot v_{exg} \cdot B_f \cdot t_{i-1} \cdot c_{exg} + \sum\limits_{k=1}^{i-1} (\Delta G_{2n-k+1} - \Delta G_k) \cdot t_{i-1} \cdot c_{exg} & \text{for} \quad i = 2, \quad \dots \quad , \quad 2n \end{cases} \tag{2}$$

$$Q_{rec\ \ i} = \begin{cases} \Delta G_{2n-i+1} \cdot t_{2n-i+1} \cdot c_{exg} & \text{for} \quad i = 1, \quad \dots \quad , \quad n/2 \quad \text{and} \quad i = n+1, \quad \dots \quad , \quad 3n/2 \\ -\Delta G_{2n-i+1} \cdot t_{2n-i+1} \cdot c_{exg} & \text{for} \quad i = n/2+1, \quad \dots \quad , \quad n \quad \text{and} \quad i = 3n/2+1, \quad \dots \quad , \quad 2n \end{cases} \tag{3}$$

$$Q_{bur\ i} = m_i \cdot B_f \cdot Q_{low} \tag{4}$$

where

$B_f$—consumption, (kg/s);

$t_f$, $t_{air}$—temperature of heated fuel and air, (°C);

$c_f$, $c_{air}$, $c_{exg}$—heat capacity of fuel, air and flue gases, (J/(m³ K));

$L_n$—actual combustion air consumption, (kg air/kg fuel);

$t$—temperature of the gases in combustion chamber, (°C);

$m_i$—fraction of fuel burned in the $i$th zone;

$v_{exg}$—specific exhaust gases output during combustion, (kg/kg).

$\Delta G_i$—consumption of gases that move between local recirculation zones, (kg/s);

$Q_{low}$—lower operating heat of combustion of fuel, (J/kg);

$n$—number of calculated zones along the length of the furnace, (pcs).

The value of $\Delta G_{ij}$ is determined by the law of the distribution of mass flow density of combustion products through the interface between the forward and reverse trajectories of their movement. For this purpose, the following formula can be used:

$$\Delta G_i = \begin{cases} 0 & \text{for} \quad 1 \leq i \leq n/2 \quad \text{and} \quad n < i \leq 3n/2 \\ A \cdot \left(\frac{2 \cdot y_i}{L_{tr}}\right)^b & \text{for} \quad n/2 < i \leq n \\ A \cdot \left(\frac{2 \cdot y_{i-n}}{L_{tr}}\right)^b & \text{for} \quad 3n/2 < i \leq 2n \end{cases} \tag{5}$$

where

$y_i = \Delta y \cdot (i - 1/2)$—coordinate of zone $i$ centre, (m);

$\Delta y = L_{tr}/(2 \cdot n)$—length of the $i$th zone, (m);

$L_{tr}$—total length gas movement trajectory, (m);

$b$—distribution function exponent;

$A$—the regularity coefficient of the distribution of the specific mass flow of combustion products across the interface of the forward and reverse trajectories of their movement, which is determined by the Equation (6):

$$B_f \cdot v_{exg} \cdot (k_p - 1) - A \cdot \left[\sum_{i=\frac{n}{2}}^{n} \left(\frac{2 \cdot y_{i+1}}{L_{tr}}\right)^b\right] = 0 \tag{6}$$

where

$k_p$—coefficient of the large scale (recirculation through all zones of the combustion chamber) recirculation.

The share of fuel that burns in the *i*th calculation zone of the furnace is determined as:

$$m_i = \begin{cases} 1 - C(L_i) & \text{for} \quad i = 1 \\ C(L_{i-1}) - C(L_i) & \text{for} \quad i = 2, \quad \ldots \quad , \quad 2n \end{cases} \tag{7}$$

where

$C(L_i) = exp\left(-\frac{K_g \cdot \Delta y \cdot i}{L_f}\right)$—the regularity of changes in the relative concentration of fuel depending on the length of the torch;

$K_g$—parameter of the fuel burnout model, which depends on the completeness of fuel combustion in the flare and the type of burner;

$L_f$—the length of the combustion zone, (m).

Output parts of the heat balance of the *i*th zone are calculated according to (8)–(10).

$$Q_{Me\ i} = C_{Me} \cdot \left[(t_i + 273)^4 - (t_{Me} + 273)^4\right] \cdot \Delta F_{Me_i} \tag{8}$$

$$Q_{lost\ i} = k_{lost} \cdot t_i \cdot \Delta F_{Me_i} \cdot \xi_{gr} \tag{9}$$

$$\begin{aligned} Q_{exg\ i} &= \left(1 - \sum_{k=1}^{i} m_k\right) \cdot \left(B_f \cdot t_i \cdot c_f + L_n \cdot B_f \cdot t_i \cdot c_{air}\right) + \\ &+ \sum_{k=1}^{i} m_k \cdot v_{exg} \cdot B_f \cdot t_i \cdot c_{exg} + \sum_{k=1}^{i} (\Delta G_{2n-k+1} - \Delta G_k) \cdot t_i \cdot c_{exg} \end{aligned} \tag{10}$$

where

$C_{Me}$—equivalent coefficient of radiation in the zone with respect to the conduction, $(W/(m^2\ K^4))$

$t_{Me}$—temperature of the material surface, (°C);

$\Delta F_{Me_i} = \frac{F_{Me}}{2 \cdot n}$—metal surface area in the *i*th zone, $(m^2)$;

$F_{Me}$—total heat exchange area of the material, $(m^2)$;

$k_{lost}$—coefficient of heat transfer to the environment, $(W/(m^2 \cdot K))$;

$\xi_{gr} = F_{lin}/F_{Me}$—gradation of furnace laying;

$F_{lin}$—total area of the furnace laying, $(m^2)$;

To determine the temperature change of the gases in the working chamber of the furnace, a system of heat balance equations was compiled for each *i*-th calculation zone. The total number of equations was $2n$. Next, by solving the system, the temperature distribution of the gases along the trajectory of their movement was relatively determined. The value $\Delta t = t_{max} - t_{min}$ was calculated as a parameter of the uniformity of gas temperature distribution in the working chamber of the furnace, where $t_{max}$ and $t_{min}$ are the maximum and minimum temperatures of gases in the furnace).

Article [9] presents a comparison of the temperature distribution in the furnace chamber with a U-shaped trajectory of flue gas movement obtained by using a mathematical model with the results of temperature distribution measurements in a physical model- laboratory furnace. It was shown that the accuracy of the calculated results was approximately 1.4–2.45%.

The mathematical model of the processes of gas dynamics and heat exchange in the heating furnace is quite complex, and does not allow establishing the analytical dependence of the uniformity of the gas temperature distribution $\Delta t$ on the parameters that affect it. The use of the method of successive approximation [3] allows establishing dependence and eliminating the need to solve a system of $2n$ nonlinear equations each time to determine the value of $\Delta t$.

### 2.2. Method of Successive Approximation

To extend the scope of the MSA application, a hypothesis was formulated, as follows.

Let there exist a scalar function $F(X) = F(x_1, x_2, x_3, \ldots x_n)$, that is bounded, defined and continuous in the closed region $\overline{D}$ of the scalar field $P$. Then for any point $M \subset \overline{D} \forall M \in D; \forall \varepsilon \geq$

$0\exists U_\varepsilon(M) \subset \overline{D}$ in the vicinity of the point $M_0\left(x_1^0, x_2^0, x_3^0, \ldots x_n^0\right)$ (Figure 2) the function $F(X)$ can be represented in the form:

$$|F(X) - \phi(X)| \leq \varepsilon \forall M_0 \in U_\varepsilon(M_0) \qquad (11)$$

where

$U_\varepsilon(M_0)$—vicinity of the point $M_0\left(x_1^0, x_2^0, x_3^0, \ldots x_n^0\right)$;
$\phi(X) = \alpha \prod_{i=1}^n g_i(x_i)$;
$g_i(x_i)$—approximation functions for $f_1, f_2, f_3, \ldots, f_n$, which are given as follows:

$$f_1(x_1) = F\left(x_1^0, x_2^0, x_3^0, \ldots, x_n^0\right), \ f_2(x_2) = F\left(x_1^0, x_2^0, x_3^0, \ldots, x_n^0\right), \ f_3(x_3) = F\left(x_1^0, x_2^0, x_3^0, \ldots, x_n^0\right), \ldots, f_n(x_n) = F\left(x_1^0, x_2^0, x_3^0, \ldots, x_n^0\right) \qquad (12)$$

$\alpha$—the approximation coefficient is determined in accordance with the formula:

$$\alpha = \frac{F(M_0)}{g_1\left(x_1^0\right) \cdot g_2\left(x_2^0\right) \cdot g_3\left(x_3^0\right) \cdot \ldots \cdot g_n\left(x_n^0\right)} \qquad (13)$$

Using the specified approach, the representation of the function $F(X) = F(x_1, x_2, x_3, \ldots x_n)$ around the point $M_0\left(x_1^0, x_2^0, x_3^0, \ldots x_n^0\right)$ has sufficient accuracy for engineering calculations on the entire definition area $\overline{D}$.

The MSA application algorithm can be represented by a sequence of the following steps:

1.  Selecting a point from the function definition area

$$M = M_0\left(x_1^0, x_2^0, x_3^0, \ldots x_n^0\right), \ M \in \overline{D} \qquad (14)$$

2.  Creating the function $f_1(x_1) = F\left(x_1^0, x_2^0, x_3^0, \ldots, x_n^0\right)$.
3.  Finding the type of function $g_1(x_1)$, that is an approximation for the function $f_1(x_1)$;
4.  Finding $\varphi_1(x_1)$ in accordance with step 1: $\varphi_1(x_1) = \alpha_1 g_1(x_1)$, where $\alpha_1$ is the approximation coefficient.
5.  Defining the function around the point $M$ from equality $F(x_1) \approx \varphi_1(x_1)$.
6.  Repeating the steps 2–5 consecutively for the variables $x_j\left(j = \overline{2, n}\right)$ and obtaining the representation as follows:

$$F(x_1, x_2, x_3, \ldots x_n) \approx \varphi_1(x_1, x_2, x_3, \ldots x_n) = \alpha \cdot g_1(x_1) \cdot g_2(x_2) \cdot g_3(x_3) \cdot \ldots \cdot g_n(x_n) \qquad (15)$$

The location of a point $M = M\left(x_1^0, x_2^0, \ldots, x_n^0\right), M \in \overline{D}$ in the definition area significantly depends on its topology, and therefore affects the way it is represented. The choice of a point on the definition area is determined by prior knowledge of its features and is determined by the researcher's experience. In the case of complex functions and lack of prior knowledge about behavior of the resulting function, it is convenient to choose it in the center of the definition area, i.e., to determine the coordinates according to the formula:

$$x_j = \frac{b_j - a_j}{2} \qquad (16)$$

where $a_j$, $b_j$ represent the beginning and end proves (respectively) of the interval of changes of parameter $x_j$.

It should be noted that the surface of the desired function is represented using a generalized hyperbolic hyperboloid, and it is not worth hoping for a uniform behavior of the relative error.

It should also be noted that the problem of choosing classes of approximation functions is one of the most critical problems, not only in applied mathematics but also in technical applications. As research has shown [12], the coefficient of variation of approximating functions cannot act as a criterion for selecting functions. As a criterion that limits the

choice of the class of approximating functions, it is suggested to choose the dimension of the original function, if possible. Using the limitations of the class of approximating functions is extremely important for evaluating the influence of parameters on the resulting function.

Wide application of MSA in practice has shown that the use of a class of power functions is particularly effective for evaluating the influence of parameters on the quality function. The impact assessment method is a reproduction of the problem solution (when it is constructed in the form of described tables of numbers) in the form of a product of power functions and a comparison of their indicators. The higher the exponent, the stronger the influence of the parameter on the function.

## 3. Initial Data, Quality Function, Independent Parameters and Their Limits

Using a mathematical model of gas dynamics and heat exchange processes, calculations of gas temperature distribution along the length of the trajectory of their movement in the working chamber of the furnace were performed.

The initial data for modeling are presented in Table 1. The most characteristic parameters of the work intended for heating steel ingots were taken as initial data. The length of the working chamber of the furnace (8 m) and height (43 m) are typical for the intended design and do not change depending on the aisles. As a rule, such furnaces are heated with a mixture of consumption of coke and blast-furnace gas with a calorific value of 5.5–812 MJ/m$^3$ [2]. At the same time, the higher the temperature of heating the fuel and air before their absorption, the lower the calorific value of the mixture of coke oven and blast-furnace gas that is used. In the assumed heating of air and visible gases, up to 800 °C and 300 °C, respectively, which is the most widely used heat leaving the flue gas furnace, the metal surface temperature of 1000 °C corresponds to the range of revolutions of its metal processing. The thermophysical properties of metal, gas, and air correspond to their temperature [10].

**Table 1.** Initial data.

| $B_f$, kg/s | $Q_f$, J/kg | $L_n$, kg/kg | $v_{exg}$, kg/kg | $t_f$, °C | $t_{air}$, °C | $t_{Me}$, °C | $c_f$, J/(kg·K) |
|---|---|---|---|---|---|---|---|
| 1.721 | $6.164 \times 10^6$ | 1.943 | 2.943 | 300 | 800 | 1000 | 1240 |
| $L_{tr}$, m | $C_{Me}$, W/(m$^2$·K$^4$) | $F_{Me}$, m$^2$ | $\xi_{gr}$ | $k_{lost}$, W/(m$^2$·K) | $n$ | $c_{air}$, J/(kg·K) | $c_{exg}$, J/(kg·K) |
| 16 | $1.731 \times 10^{-8}$ | 122 | 1.344 | 2 | 8 | 1071 | 1238.2 |

As a quality function, an indicator that characterizes the uniformity of the gas temperature in the working chamber $F = t_{max} - t_{min}$ was chosen, where $t_{max}$ and $t_{min}$ are the maximum and minimum gas temperatures along the trajectory of their movement, respectively. According to the results of previous studies, six independent parameters that affect the quality function were determined, and the range of their possible change was also established.

(a) Fuel temperature $t_f = 0, \dots, 300$ °C. It is known from literature sources [13] that it is not safe to increase the heating temperature of gaseous fuel in the existing heat exchangers of furnaces above 300 °C, for the reasons of explosive safety.

(b) The temperature of the air used for fuel combustion is $t_{air} = 0, \dots, 1000$ °C. Due to the heating of air to high temperatures, significant fuel savings are achieved in methodical [14], ring [15], roller and shaft [16] furnaces, as well as in heating wells [17] of various designs. The maximum air heating temperature depends on the method of utilization of the heat of the flue gases leaving the furnace and the design of the heat exchangers, but usually this temperature does not exceed 1000 °C. At the same time, in outdated equipment that is still in operation at some enterprises, heating of the combustion air before burning is not provided.

(c) Multiplicity of internal large-scale recirculation of furnace gases $k_p = 1, \dots, 6$.

The choice of such limits for varying the multiplicity of recirculation of furnace gases is determined by the practical research data obtained by numerous experimenters. So, when modeling a recirculation circuit with a built-in fan, V.D. Brook [18] discovered that increasing the recirculation ratio above 5 was impractical and led to an increase in fan power consumption. A shown in research conducted at the Gas Institute of the National Academy of Sciences [19], when the recirculation rate was increased above $k_p = 6$, the intensity of temperature equalization in the furnace decreased sharply, so further intensification of recirculation is impractical.

The indicator of the degree of the distribution function, which characterizes small-scale recirculation, was chosen on the base of considerations outlined in the paper [4] within $b = 0, \ldots, 15$.

The parameter of the fuel burnout model adopted by V.Y. Gubynskyi et al. [2] varies within the limits $K_g = 3, \ldots, 4.61$;

The length of the combustion zone $L_f = 8, \ldots, 16$ m was taken in metres from the conditions that the torch occupies the entire length of the furnace space, and according to the operation data of the operating furnaces.

Acting in accordance with the MPA algorithm, the range of change for each independent parameter was divided into 10 intervals with equal steps and a reference point was chosen $M\left(t_f^0, t_{air}^0, k_p^0, b^0, K_g^0, L_f^0\right)$, where $t_f^0 = 150\,°C$; $t_{air}^0 = 500\,°C$; $k_p^0 = 3.5$; $K_g^0 = 3.805$; $L_f^0 = 12$ m.

## 4. Results

Figure 5 shows the results of calculating the gas temperature distribution in the working chamber of the furnace. As can be seen from the figure, a gas minimal temperature value of 1053.9 °C was observed at the entrance to the working chamber of the furnace. The maximal gas temperature was observed in the sixth calculation zone, reaching 1268.4 °C. The quality function value at the reference point was $F(t_f^0, t_{air}^0, k_p^0, b^0, K_g^0, L_f^0) = 214.5\,°C$.

Figures 6–11 show combined graphs of the function $F$ values (marked by dots in the figures) and their approximations functions $g_i$ (solid continuous lines).

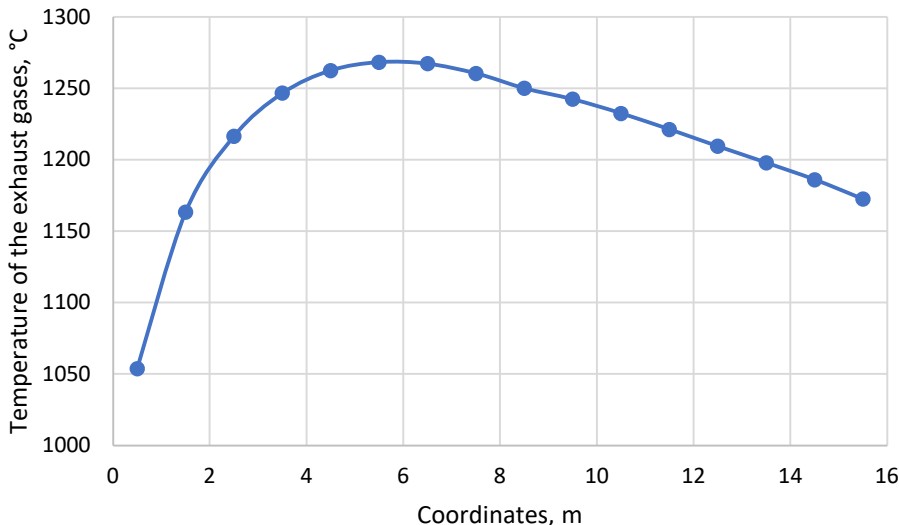

**Figure 5.** Distribution of gas temperature along the trajectory of movement.

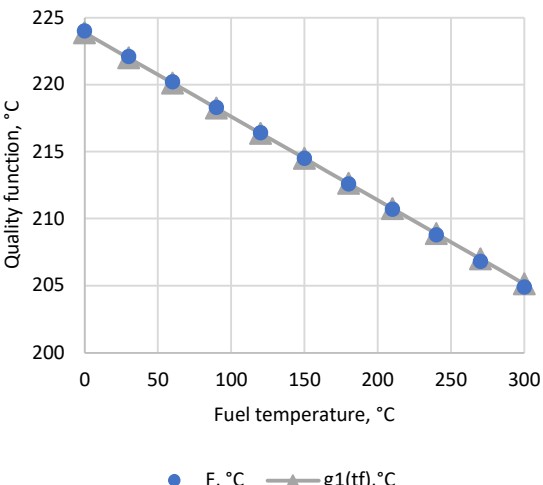

**Figure 6.** Combined graph of the values of the function $F$ from the parameter $t_f$.

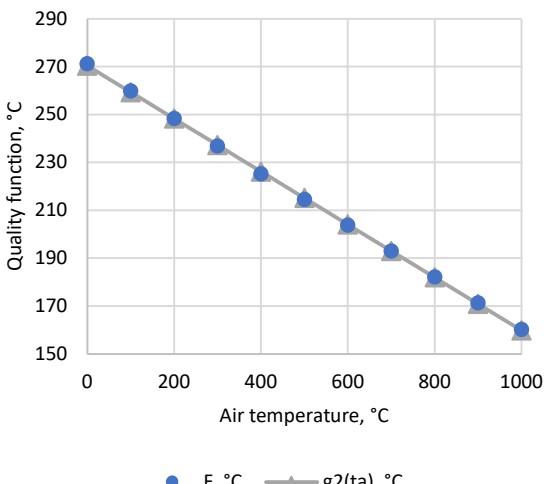

**Figure 7.** Combined graph of the values of the function $F$ from the parameter $t_{air}$.

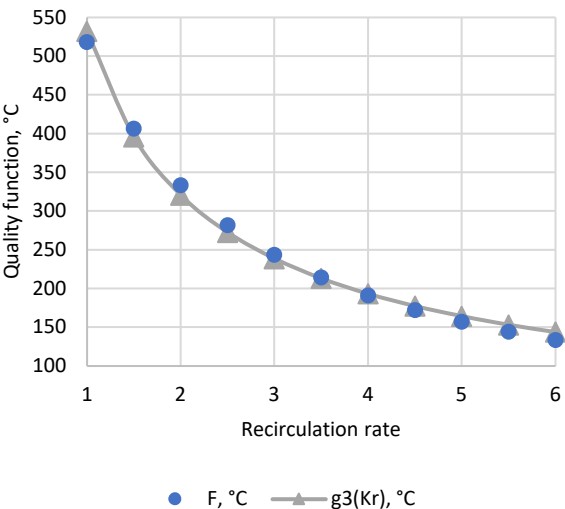

**Figure 8.** Combined graph of the values of the function $F$ from the parameter $k_r$.

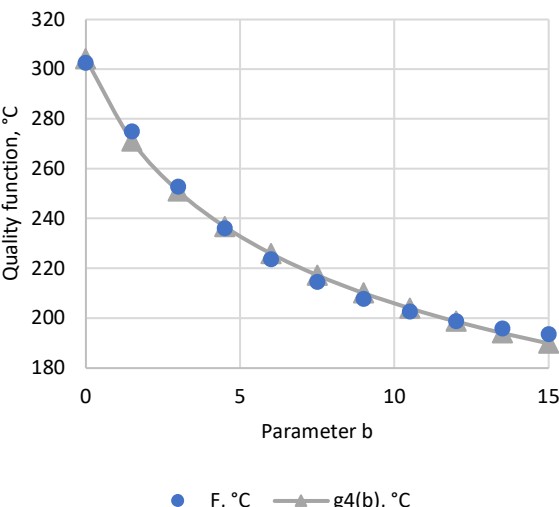

**Figure 9.** Combined graph of the values of the function *F* from the parameter *b*.

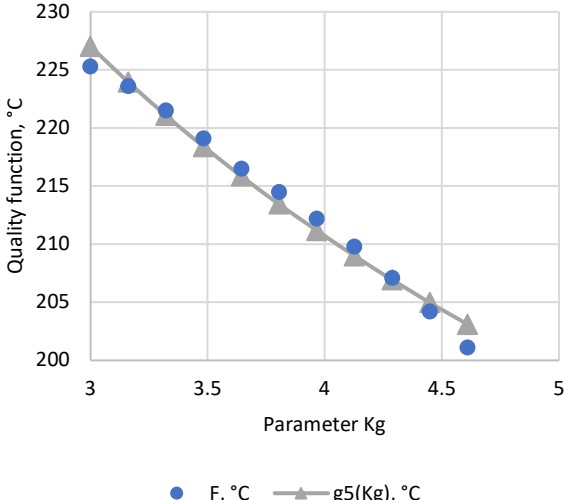

**Figure 10.** Combined graph of the values of the function *F* from the parameter $K_g$.

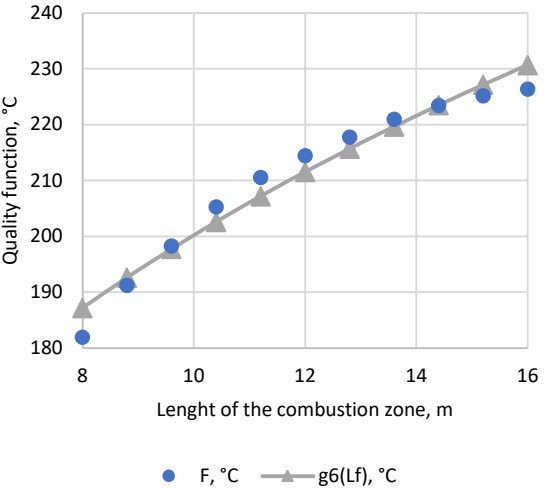

**Figure 11.** Combined graph of the values of the function *F* from the parameter $L_f$.

The approximation functions reached the following values:

$$g_1 = \left(223.75 - 0.0623939\, t_f\right) \quad g_2 = (270.441 - 0.110591\, t_{air})$$
$$g_3 = 532.222/k_p^{0.729813} \quad g_4 = \frac{376.11426000}{(b+2.43204800)^{0.23907803}} \tag{17}$$
$$g_5 = 301.811/K_g^{0.259232} \quad g_6 = 100.019 L_f^{0.301486}$$

The choice of approximating functions $g_i$ was carried out based on the best fit to the data obtained. Despite the above-mentioned advantages of approximating the obtained data by power functions, their use is not always possible. So, according to Figures 5–7, the dependencies of the quality function on air and fuel temperatures cannot be approximated by power functions, since they are linear dependencies [2]. Therefore, an exception was made for $g_1$ and $g_2$. The approximation function $g_4$ took on this form solely because of physical considerations and conditions of existence in the denominator of the expression close to zero, which leads to uncertainty. All other functions were chosen in the form of exponents for the convenience of considering the sensitivity of the function to the parameters.

According to the above hypothesis and algorithm, the calculated function $F$ takes the form:

$$F = \frac{0.0134281171 \cdot L_f^{0.301486}\left(223.75 - 0.0623939 \cdot t_f\right)(270.441 - 0.110591 \cdot t_{air})}{(b + 2.432048)^{0.23907803} \cdot K_g^{0.259232} \cdot k_r^{0.729813}} \tag{18}$$

The relative errors of the approximating functions $g_1, g_2, g_3, g_4, g_5, g_6$ are characterized by graphical dependencies presented in Figure 12.

Figure 12 presents the distribution of relative deviations of the function from the one obtained by the formula on the lines of intersection of the functional space with coordinate planes. The accuracy obtained by the choice of approximating functions is at a satisfactory level for most of engineering calculations because the obtained relative errors are commensurate with the measurement errors of the corresponding quantities in industrial conditions. The behavior of errors on the hyper diagonal of the domain of definition is of interest. According to the authors, these deviations are decisive for evaluating the behavior of the function and adjusting the scope of its definition, that is, in fact, the range of the parameter variations.

As can be seen from the above graph, the last error value (32.8%) is not desirable for engineering calculations. So, for example, with the temperature distribution shown in Figure 6, the value of the quality function is 214.5 °C. With an error value of 32.8%, the error in determining the temperature unevenness in the working chamber of the furnace reaches 70.4 °C, which significantly exceeds the error in measuring temperatures in industrial conditions (10–15 °C).

To reduce the relative error when using the given formula, it is necessary to reduce the domain of the function by reducing the right-hand interval by the step of dividing the range of variations. The reduction is not significant but allows the range of errors in the definition domain of the function to be limited to 6%, which is a satisfactory result for most engineering calculations.

Figure 14 shows the graph of the relative error for the adjusted range of parameters: fuel temperature $t_f = 0, \ldots, 270$ °C; the temperature of the air used for fuel combustion $t_{air} = 0, \ldots, 900$ °C; multiplicity of internal large-scale recirculation of furnace gases $k_r = 1, \ldots, 5.5$; an indicator of the degree of distribution function characterizing small-scale recirculation $b = 0, \ldots, 13.5$; a parameter of the fuel burnout model $K_g = 3, \ldots, 4.449$, and the length of the combustion zone $L_f = 8, \ldots, 15.2$ m. In the specified range of parameter variations, the relative error of the quality function does not exceed 6%.

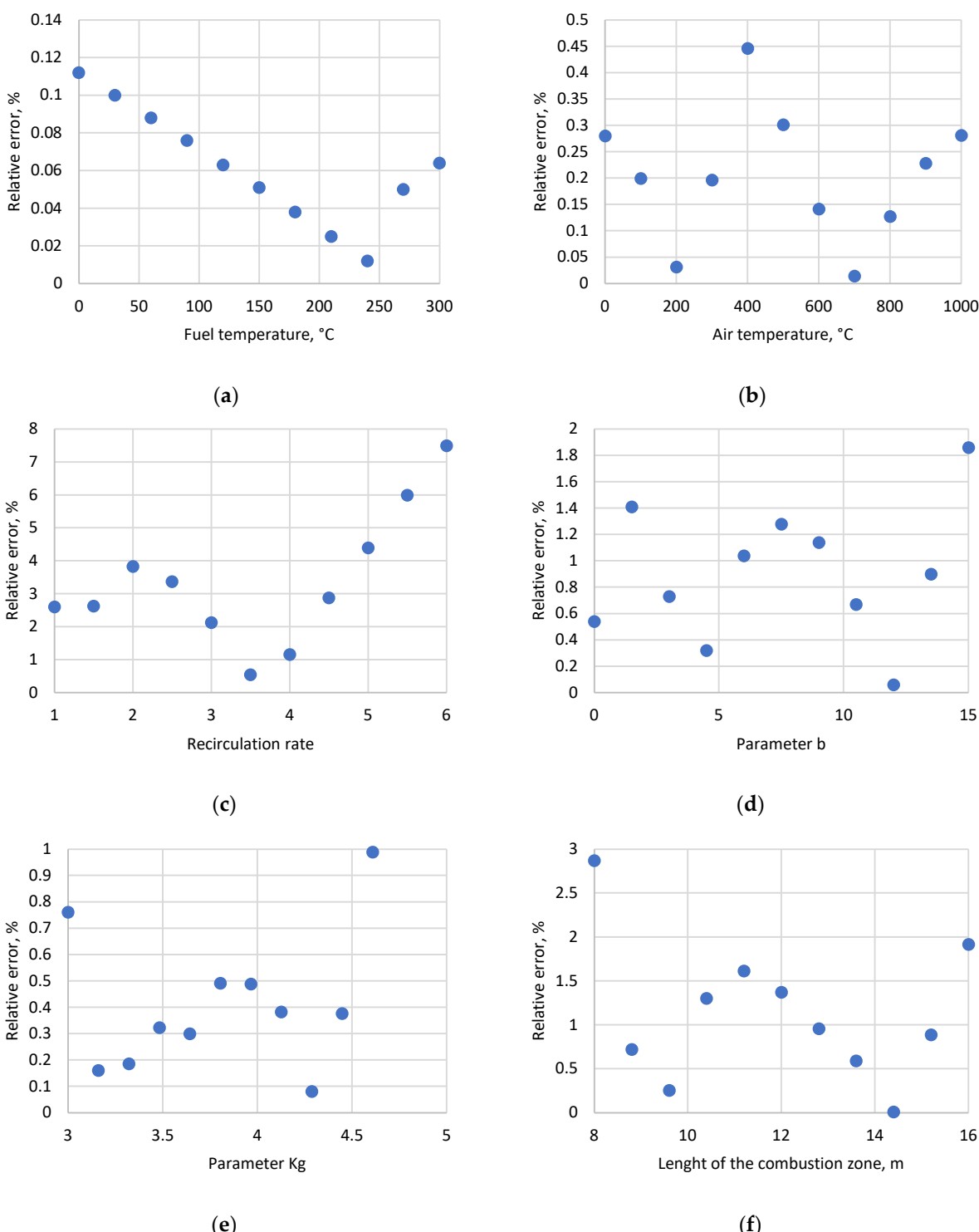

**Figure 12.** Relative error of approximating functions: (**a**) $g_1$, (**b**) $g_2$, (**c**) $g_3$, (**d**) $g_4$, (**e**) $g_5$, (**f**) $g_6$.

The distribution of the relative error on the generalized diagonal of the hypercube of the definition area is presented in Figure 13.

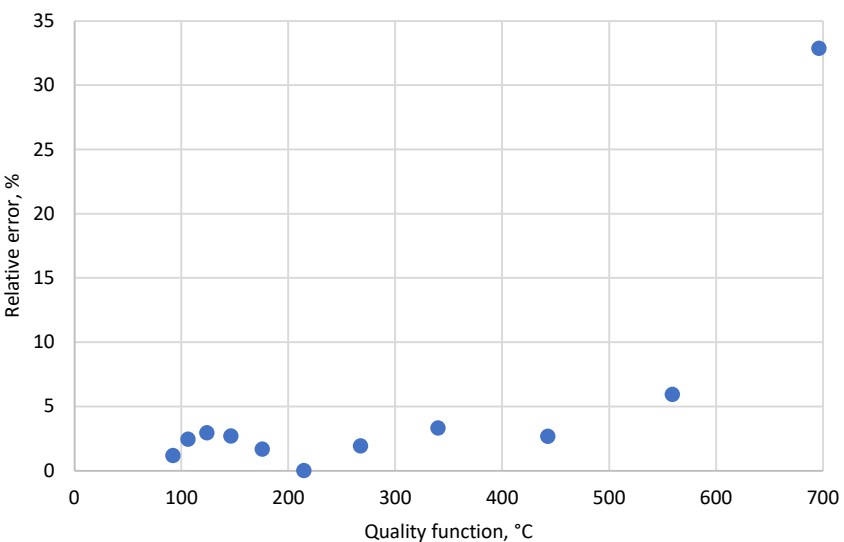

**Figure 13.** Relative error on the diagonal of the definition area.

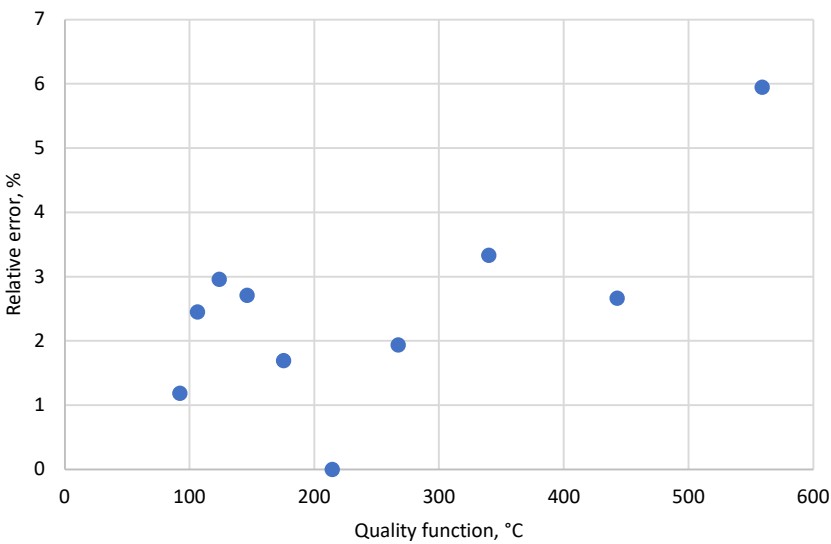

**Figure 14.** Relative error on the diagonal of the definition area of the parameters $t_f$, $t_{air}$, $k_r$, $b$, $K_g$, $L_f$.

## 5. Discussion

Based on the hypothesis formulated in previous section, the results can be written as follows:

$$f_1^*(x_1, x_2) = F^*(x_1, x_2, x_3^0, x_4^0, \ldots, x_n^0) = F(M_0)/(g_1(x_1^0)\, g_2(x_2^0))$$
$$f_2^*(x_1, x_2,\, x_3) = F^*(x_1, x_2, x_3, x_4^0, \ldots, x_n^0) = F(M_0)/(g_1(x_1^0)\, g_2(x_2^0)g_3(x_3^0))$$
$$f_3^*(x_1, x_2,\, x_3, x_4) = F^*(x_1, x_2, x_3, x_4, x_5^0, \ldots, x_n^0) = F(M_0)/(g_1(x_1^0)\, g_2(x_2^0)g_3(x_3^0)g_4(x_4^0)),$$
$$f_4^*(x_1, x_2,\, x_3, x_4,\, \ldots,\, x_n) = F^*(x_1, x_2, x_3, x_4, x_5, \ldots, x_n) = F(M_0)/(g_1(x_1^0)\, g_2(x_2^0)g_3(x_3^0)\ldots g_n(x_n^0)).$$

That means that moving sequentially, starting from the first step, one can obtain the entire sequence of representations of the function from these parameters. It all works when all other points are at the point $M_0\left(x_1^0, x_2^0, x_3^0, \ldots x_n^0\right)$.

The simulation results given above fully reflect qualitative ideas about furnace processes, which are known from their operation. Thus, among the selected independent parameters, the most significant influence on the uniformity of the temperature field in furnaces with a U-shaped trajectory of gas movement is characterized by the multiplicity of large-scale recirculation $k_r$. Changing this indicator value from $k_r = 1$ to $k_r = 6$ results in a tenfold decrease of the quality function value $F = t_{max} - t_{min}$ meaning the uniformity

of the gas temperature in the working chamber of the furnace. The simulation results confirmed the thesis of a sharp drop in the intensity of temperature equalization with an increase in the recirculation rate. The processing of the approximation curve of the quality function *F* shows that the uniformity of the gas temperature increases by almost 40 % when the recirculation ratio increases from $k_r = 1$ to $k_r = 2$ and by 12.5% only when the same parameter increases from $k_r = 5$ to $k_r = 6$.

Small-scale recirculation has a similar quality, but smaller influence on function *F*. Such an effect on temperature uniformity in the working chamber of the furnace is quite natural. After all, both small-scale and large-scale recirculation characterize gas-dynamic and mass exchange processes, which are described by the same equations and have the same effect on the heat exchange. Analysis of the data on the impact of small-scale recirculation shows that with an increase in its share, the unevenness of the gas temperature distribution in the furnace can increase by an additional 15%. The reduction of small-scale local recirculation is associated with the elimination of obstacles and local resistances in the path of movement of furnace gases, and a sharp change in the direction of their movement.

The gas heating temperature and the parameters of the fuel combustion model have the least influence on the uniformity of the gas temperature out of the six selected parameters. The change in fuel temperature from the minimum to the maximum level (i.e., from 0 °C to 300 °C) leads to an insignificant improvement of the uniformity of gas temperature distribution in the working chamber of the furnace, as it ranges from 225 °C to 205 °C. The parameter of the model of gas combustion in the torch has almost the same effect.

The length of the flare affects the quality function *F* in direct proportion. Studies have confirmed the growth of heating unevenness with increasing torch length. In our case, such an increase value was 45 °C, which means an almost 25% increase.

As follows from Figure 12, the relative errors of the approximating functions $g_1$, $g_2$, $g_3$, $g_4$, $g_5$, $g_6$ are not significant and do not exceed 2–3% except for the error of the approximating function $g_3$ (multiplicity of recirculation) which reaches 7.5% at the limit point equal to 6. Under the conditions where such a value of the recirculation multiplicity for furnaces is too large, it may be considered that the accuracy of the approximation functions obtained in research is quite acceptable for engineering calculations.

The same conclusion can be made about the value of the relative error on the diagonal of the detection area on the base of Figure 13. Except for the extreme point, the relative error does not exceed 6%, which is within the limits for most of engineering calculations and measurement errors when using industrial metrological devices.

According to the literature [20,21], MSA is a suitable and effective method of mathematical modeling and calculation. The results of the studies carried out in this work indicate the possibility of expanding the scope of MSA to solve gas dynamics and heat transfer problems.

## 6. Conclusions

This paper presents the results of applying the successful approximation method for mathematical modeling of gas dynamics and heat transfer processes in a heating furnace with a U-shaped trajectory of flue gas movement. The influence of a number of independent parameters on the selected quality function, i.e., the uniformity of the temperature field of gases in the working chamber of the furnace, was studied. The following parameters were selected as independent parameters: fuel and air heating temperatures before combustion; the multiplicity of internal large-scale recirculation of furnace gases; the exponent of the distribution function, which characterizes the small-scale recirculation of gases in the working chamber; the flame burnup model parameter, and the length of the burning zone. It was established that the greatest influence on the quality function is exerted by the multiplicity of the internal large-scale recirculation of furnace gases, the change of which from 1 to 6 leads to a tenfold increase in the uniformity of the gas temperature distribution in the working chamber. Moreover, the main increase in the uniformity of temperature distribution (up to 40%) occurs with an increase in the recirculation ratio from 1 to 2. With a

further increase in the recirculation ratio, the effect of this parameter on the quality function weakens and does not exceed 12.5% with a change in the recirculation ratio from 5 to 6. The fuel heating temperature before combustion has the least effect on the quality function (up to 8.9%). The remaining parameters considered are characterized by an average influence on the quality function. Their change in the considered interval causes a change in the uniformity of the temperature field of gases by 15–25%.

MSA, proposed for the approximate determination of regularities between system parameters, was confirmed for its efficiency in the calculation of gas dynamics and heat transfer problems. Relative errors for most parameters did not exceed 6%, which is a confirmation of the efficiency of the algorithm, on the one hand, and the possibility of using simplified formulas for practical engineering calculations.

The simplified method proposed by the authors, in contrast to that used in the aforementioned works, does not require continuity and smoothness of functions and does not use derivatives of the quality function, which widely extends the scope of its application in various areas of science including mathematical modeling of gas dynamic and heat transfer processes in industrial heating and thermal equipment.

The representation of the product functions by power functions allows swift evaluation of their influence on the quality function. Our research also indicates the possibility of using MSA as a tool for a digital twin design for thermal processes. In the future, the authors aim to continue with mathematical and experimental verification of the presented hypothesis.

**Author Contributions:** Conceptualization, M.R. and O.Y.; methodology, O.G.; software, A.K.; validation, G.L., O.G., Y.M. and A.K.; formal analysis, M.F.; investigation, O.Y.; resources, M.R.; data curation, A.K. and T.K.; writing—original draft preparation, O.G. and A.K.; writing—review and editing, A.K. and T.K.; visualization, A.K.; supervision, O.Y.; project administration, M.R.; funding acquisition, M.R. All authors have read and agreed to the published version of the manuscript.

**Funding:** The work is supported by KEGA grant agency under Grant KEGA 023TUKE-4/2021.

**Institutional Review Board Statement:** Not applicable.

**Informed Consent Statement:** Not applicable.

**Data Availability Statement:** Not applicable.

**Conflicts of Interest:** The authors declare no conflict of interest.

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
