# Peer review of "Method of Sequential Approximation in Modelling the Processes of Heat Transfer and Gas Dynamics in Combustion Equipment"

_applsci, doi:10.3390/app122311948_

Round 1

Reviewer 1 Report

Review of manuscript No applsci-2020276

Method of Sequential Approximation in Modelling the Processes of Heat Transfer and Gas Dynamics in the Combustion Equipment by Miroslav Rimar, Oleksandr Yeromin, Grigoriy Larionov, Andrii Kulikov, Marcel Fedak, Tibor Krenicky *, Olena Gupalo and Yana Myanovskaya

The title of the draft manuscript is informative about the research paper. The main aim of the research paper has been defined clearly. The methodology of research study is easy to follow. Moreover, it seems that the manuscript’s purpose is to show that the applied tool behave as expected. The computational results are presented appropriately. The depicted results are placed into context without being over interpreted. For more contribution, the Authors compared their results with those in relevant published works of other researchers. The manuscript's English is generally not bad. In general, whole manuscript is almost well organized in chapters. Furthermore, the current work is well documented in the research field (21 references). Nevertheless, I do not recommend the draft manuscript in the present version for publication in this journal. Minor points in the article which needs clarification, refinement and/or additional information and suggestions are listed below:

1/ The abstract reads very general. Please include some quantitative data related to research outcomes.

2/ Figure 1 is illegible and needs improvements in order to be more readable. Reviewer suggests rebuild of algorithm scheme.

3/ The results might be relevant for the specific system, but the question is what is the added value for other researchers?

4/ It is a wrong practice to use the equations, without mentioning the source.

5/ Input data used to modeling should be presented in separately section, before section 3 Results.

6/ Why did you choose these input data for the modelling? Any advantageous, limits?

7/ There is an typo in the description of OY axis. The same problem goes for Fig. 15.

8/ Due to the fact that this work concerns numerical investigation, Reviewer did not find the reply to question: What error formula was used to estimate rightness of calculations?

9/ Unjustified conclusion (see page 5 lines 449-450). There is no comparative analysis to other methods, e.g. fuzzy logic.

10/ The key findings presented in the conclusion do not surprisingly include any quantitative results, like in the abstract. Those parts should be revised. That is, the key findings should be provided with supported quantitative data.

11/ Please provide recommendations to future works in this research field. I suggest that future research and lines of research should be better listed and highlighted at the end of conclusions section.

On the basis of above-mentioned considerations, major revision of draft manuscript is suggested before resubmit to Applied Sciences.

Author Response

Dear Reviewer, we would like to thank you for your precise reading of the manuscript and highly professional comments aimed at its improvement. We tried to do our best to revise the manuscript to meet your proposals and comments. For more details, please see the attached file and manuscript.

Reviewer 2 Report

This paper investigates the application of the method of successive approximation for gas dynamics and heat transfer processes in furnaces. I have the following comments:

ABSTRACT: The second part on the details of the explanation of the method is not clear. I would suggest avoiding explanation of the method here, and to reformulate this part to give a more precise overview of the main message, including quantitative information on the results obtained in the paper

KEYWORDS: I would suggest avoiding acronyms, e.g. MSA

INTRODUCTION: this part appears to be more similar to methods that a state of the art. It is also very general, and the advancement on the literature gap on the object of study is not sufficiently clear. I would suggest moving the general explanation of the research algorithm and the related figures to “Materials and methods”, and to improve the introduction with a more quantitative discussion and solid referencing to the limits of the simulations, specifically for the case considered in this study.   

SECTION 2.1: In view of the previous comment, the header could be changed to “Test case” or similar so that, after the general algorithm is introduced, the attention is focused on the considered application  

SECTION 2.1: It is stated that the drawback of furnaces with a U-shaped trajectory is the non-uniformity of gas temperatures in the working chamber. It is not clear why this type of furnace is chosen as the test case: is it one of the most common layouts? Is there any particular interest to justify this layout? Are there other layouts available with better performance?

SECTION 2.1: explanation of the variables and parameters appearing in the equations should not be itemized but explained in the text below equations. Given the large quantity of labels, maybe a nomenclature section could also be added at the end of the paper, depending on the author preferences

SECTION 3: The initial data for the calculations of gas temperature distribution along the length of the trajectory should be resumed into a Table.

SECTION 3: I would suggest avoiding referencing as “Professor V.Y. Gubynskyi [2]”. This is not a standard scientific referencing format. Please use, e.g. “Gubynskyi et al. [2]”

SECTION 3: Regarding Figs. 7-12, it is stated that: “The selection of approximation functions, in addition to the best fit to the obtained data, was also influenced by the fit with physical ideas about the nature of the processes being studied.” The referred nature of each process being studied should be explained

SECTION 3: Regarding Figs. 7-12, it is stated that: “Thus, the functions ?1 and ?2 are chosen to be linear, despite the wishes to find them in the form of power functions, due to the compatibility of the processes under consideration.” This inconsistency and the “compatibility of the processes under consideration” should be explained, with supporting references.

SECTION 3: Regarding Fig. 13, it is stated that “The accuracy, which is at a satisfactory level for most of engineering calculations…”. The different trends and degree of qualitative recovery of the data points should be discussed in relation of the physical meaning of the different parameters

SECTION 3: Regarding Fig. 14, it is stated that: “the last error value is not desirable for engineering calculations”. This statement should be justified with a practical example

Author Response

Dear Reviewer, we would like to thank you for your precise reading of the manuscript and highly professional comments aimed at its improvement. We tried to do our best to change the manuscript to meet your proposals and comments. For more detailed answers, please see the attached file and manuscript.

Round 2

Reviewer 1 Report

Reviewer has re-examined the manuscript. The draft manuscript still needs improvements. Authors should be considered the following comment/questions in order to revise the manuscript.

1/ Fig. 12 needs improvements, especially the description of OY axis. There are a lot of typos. Typing suggestions: "relative eror" => "relative error". The same problem goes for Figs. 13 and 14.

2/ How obtained results were validated?

3/ What does mean dotted curves in Figs. 12, 13 and 14?

Authors should consider above-mentioned remarks in order to revise the draft manuscript. Reviewer thinks that a publication of the draft manuscript may be possible after minor revision

Author Response

Dear Reviewer, we would like to thank you once again for your precise reading of the manuscript and highly professional comments aimed at its improvement. We tried to do our best to change the manuscript to meet all your proposals and comments.
